# Targeting Aberrant FGFR Signaling to Overcome CDK4/6 Inhibitor Resistance in Breast Cancer

**DOI:** 10.3390/cells10020293

**Published:** 2021-02-01

**Authors:** Navid Sobhani, Anne Fassl, Giuseppina Mondani, Daniele Generali, Tobias Otto

**Affiliations:** 1Department of Medicine, Section of Epidemiology and Population Sciences, Baylor College of Medicine, Houston, TX 77030, USA; 2Department of Cancer Biology, Dana-Farber Cancer Institute, Boston, MA 02215, USA; anne_fassl@dfci.harvard.edu; 3Department of Genetics, Blavatnik Institute, Harvard Medical School, Boston, MA 02115, USA; 4Department Breast Oncoplastic Surgery Royal Cornwall Hospital, Treliske, Truro TR13LJ, UK; gmcorno@outlook.com; 5Department of Medical, Surgical and Health Sciences, University of Trieste, Cattinara Hospital, 34149 Trieste, Italy; dgenerali@units.it; 6Department of Internal Medicine III, University Hospital RWTH Aachen, Pauwelsstrasse 30, 52074 Aachen, Germany

**Keywords:** breast cancer, FGFR, CDK4, CDK6, inhibitor, therapy resistance

## Abstract

Breast cancer (BC) is the most common cause of cancer-related death in women worldwide. Therapies targeting molecular pathways altered in BC had significantly enhanced treatment options for BC over the last decades, which ultimately improved the lives of millions of women worldwide. Among various molecular pathways accruing substantial interest for the development of targeted therapies are cyclin-dependent kinases (CDKs)—in particular, the two closely related members CDK4 and CDK6. CDK4/6 inhibitors indirectly trigger the dephosphorylation of retinoblastoma tumor suppressor protein by blocking CDK4/6, thereby blocking the cell cycle transition from the G1 to S phase. Although the CDK4/6 inhibitors abemaciclib, palbociclib, and ribociclib gained FDA approval for the treatment of hormone receptor (HR)-positive, human epidermal growth factor receptor 2 (HER2)-negative BC as they significantly improved progression-free survival (PFS) in randomized clinical trials, regrettably, some patients showed resistance to these therapies. Though multiple molecular pathways could be mechanistically responsible for CDK4/6 inhibitor therapy resistance, one of the most predominant ones seems to be the fibroblast growth factor receptor (FGFR) pathway. FGFRs are involved in many aspects of cancer formation, such as cell proliferation, differentiation, and growth. Importantly, FGFRs are frequently mutated in BC, and their overexpression and/or hyperactivation correlates with CDK4/6 inhibitor resistance and shortened PFS in BC. Intriguingly, the inhibition of aberrant FGFR activity is capable of reversing the resistance to CDK4/6 inhibitors. This review summarizes the molecular background of FGFR signaling and discusses the role of aberrant FGFR signaling during cancer development in general and during the development of CDK4/6 inhibitor resistance in BC in particular, together with other possible mechanisms for resistance to CDK4/6 inhibitors. Subsequently, future directions on novel therapeutic strategies targeting FGFR signaling to overcome such resistance during BC treatment will be further debated.

## 1. Introduction

Breast cancer (BC) is the most common cancer among females worldwide, with an estimated incidence rate of 279,100 new cases per year just in the United States, and the leading cause of cancer death in women worldwide [1,2]. In spite of substantial improvements in diagnosis and treatment, breast cancer remains a significant global health burden. Ten percent of newly diagnosed patients have locally advanced or metastatic disease. Additionally, up to 30% of patients diagnosed with early breast cancer eventually relapse and develop metastatic disease [3]. Approximately 70% of breast cancers express the estrogen receptor (ER), and estrogen signaling drives breast cancer cell growth and progression [2]. Endocrine therapies are commonly used to treat ER^+^ breast cancer, including tamoxifen, aromatase inhibitors, and the selective estrogen receptor degrader fulvestrant. Although these endocrine therapies improved the survival of ER^+^ breast cancer patients, a large proportion of patients with metastatic BC eventually acquire resistance to endocrine therapy [4].

Recently, a combination of inhibitors of cyclin-dependent kinases 4 and 6 (CDK4/6)—palbociclib, ribociclib, and abemaciclib—with aromatase inhibitors or the ER inhibitor fulvestrant resulted in a markedly improved progression-free survival (PFS) compared to endocrine therapy alone in patients with advanced ER^+^ breast cancer [5]. Even though the majority of patients with advanced disease treated with CDK4/6 inhibitors and antiestrogens benefit from this combination, virtually all patients eventually display disease progression, underscoring the need to discover mechanisms of resistance to this new standard of care. Preclinical evidence evinced various mechanisms contributing to the intrinsic or acquired resistance to CDK4/6 inhibitors, and predominantly, among these are alterations in the fibroblast growth factor receptor (FGFR) signaling pathway. For these reasons, the focus of this review is to discuss the role of *FGFR* alterations during the development of resistance to CDK4/6 inhibitors and how targeting the FGFR pathway could be utilized therapeutically to overcome CDK4/6 inhibitor therapy resistance in BC will be further debated.

## 2. Biology of FGFR Signaling

Fibroblast growth factor receptors constitute a family of four single-pass transmembrane receptors (FGFR1–4) with an intracellular tyrosine kinase domain. They become activated upon the extracellular binding of their cognate ligands from the fibroblast growth factor (FGF) family. To this end, their extracellular parts contain three immunoglobulin (Ig)-like domains (IgI-IgIII), as well as an acidic region between IgI and IgII. While IgI and the acidic region seem to play an inhibitory role, the IgII and IgIII domains cooperate in ligand binding [6]. Interestingly, alternative splicing of the third Ig-like domain (IgIII) in FGFR1–3 yields two additional isoforms, namely IIIb and IIIc, which are mainly expressed in epithelial and mesenchymal tissues, respectively [7,8]. Similarly, the functional relevance of distinct family members is tissue-specific to some extent, with FGFR1 and FGFR3 being more important in mesenchymal tissues and FGFR2 being more relevant in epithelial tissues [9]. The diversity of receptor structures yields receptors with different binding affinities for the plethora of FGF ligands [10,11]. In addition, a fifth FGFR-like protein has been identified (FGFR5/FGFRL1), which can also bind FGF ligands but lacks the tyrosine kinase domain essential for intracellular signaling [12]. Nevertheless, FGFR5 may be involved in FGFR signaling by functioning as a coreceptor for FGFR1 [13].

While the nature of the ligand, as well as tissue-specific expression and splicing of the four FGFR receptor tyrosine kinases determine the precise cellular outcome of FGFR signaling, the activation of intracellular signaling cascades frequently impinge on the pathways involved in proliferation, cellular survival, and differentiation. Thereby, FGF-FGFR signaling mediates multiple physiological processes such as organ development, tissue homeostasis, metabolism, wound repair, and angiogenesis.

Since most of the specificity of FGF-FGFR signaling can be attributed to the nature of the ligand, it is not surprising that the FGF family comprises a large and functionally diverse group of proteins. In humans, this family consists of 22 members and can be divided into three major groups. First, the group of 15 canonical (or paracrine) FGF ligands acts in an autocrine or paracrine fashion by binding to FGFRs in complex with extracellular heparan sulfate proteoglycans (HSPGs). HSPGs are important for this signaling, since they protect FGFs from degradation and stabilize the interaction between FGFs and FGFRs. Canonical FGF ligands can be further divided into five subfamilies, i.e., the FGF1/2, FGF4/5/6, FGF3/7/10/22, FGF8/17/18, and FGF9/16/20 subfamilies. Second, the group of endocrine FGF ligands (FGF19/21/23) also binds to FGFRs but utilizes Klotho or β-Klotho as coreceptors and does not bind to HSPGs. Therefore, they are capable of diffusing into the bloodstream and act as hormones. Most notably, they regulate bile acid synthesis, glucose, and lipid metabolism, as well as vitamin D and phosphate levels. Fgf15 is also a member of this group but exists only in mice and is a homolog to human FGF19 [14]. Third, the group of intracrine (or intracellular) FGFs (FGF11–14) do not bind FGFRs. Instead, they perform intracellular functions in the nervous system [15].

FGF ligands activate FGFR signaling by triggering FGFR dimerization and the trans-phosphorylation of intracellular tyrosine kinase domains (Figure 1). Specific phospho-tyrosine residues then provide docking sites for multiple adapter proteins that induce downstream signaling cascades. Among them, FGFR substrate 2 (FRS2) is probably one of the most important mediators, although it regulates only a subset of FGFR functions [16]. Upon binding to the juxtamembrane intracellular domain of FGFR, FRS2 (as well as FRS3) becomes phosphorylated at multiple tyrosine residues, thereby recruiting the SHB-SHP2 complex and, subsequently, the GRB2-SOS complex, which, in turn, activates RAS and the well-known ERK1/2 MAPK signaling cascade [17]. In addition, ERK1/2 can become activated in a RAS-independent manner via the recruitment of members of the CRK protein family to activate FGFR [18]. Although the ERK1/2 MAPK cascade is not the only signaling pathway activated by FGFRs, it mediates many FGFR functions in diverse biological contexts, and FGFR signaling is a major driver of ERK1/2 activation during development [9]. Depending on the cellular context and the nature of the ligand, FGFRs can also activate a number of additional pathways. For instance, the adapter protein FRS2 not only activates ERK1/2 but, also, induces the PI3K-AKT signaling pathway via the recruitment of GRB2-associated protein 1 (GAB1) [19], e.g., during neuronal development or in lens cells [9]. Furthermore, FGFRs can directly recruit phospholipase C γ (PLCγ) to their autophosphorylated carboxy terminus, leading to PLCγ phosphorylation, thereby activating protein kinase C (PKC) [20]. Moreover, FGFRs were shown to activate the JAK-STAT signaling pathway. For instance, FGF1-FGFR3 recruits and activates STAT1 to inhibit proliferation in chondrocytes [21]. Intriguingly, this pathway is also exploited in cancer. In this regard, the amplification of *FGFR1* in cancer cells was shown to trigger the recruitment and activation of STAT3 [22]. Similarly, a single-nucleotide polymorphism (SNP) in *FGFR4* (rs351855, causing G388R substitution) is associated with many types of cancers and increases the recruitment and activation of STAT3, thereby enhancing cancer progression [23]. Finally, other MAPK signaling pathways can become activated upon FGFR signaling. In hepatocytes, for instance, FGF19-FGFR4 induces activation of the JNK signaling pathway, which results in reduced bile acid synthesis via repression of the critical enzyme CYP7A1 [24]. Furthermore, in chondrocytes, FGF1 and FGF18 were shown to activate p38 MAPK signaling [25,26].

Another aspect of FGFR signaling is the mechanisms involved in the termination and negative regulation of this signaling cascade. In this regard, the ubiquitin ligase CBL plays an important role. Upon the activation of FGFRs, FRS2 and GRB2 participate in recruiting CBL. CBL then ubiquitinates FGFRs and FRS2, thereby triggering endocytosis and the proteolytic degradation of FGFRs [27]. Furthermore, CBL is also involved in recruiting Sprouty proteins to activated FGFRs, leading to the inhibition of downstream ERK1/2 signaling [28]. In addition, GRB14 was suggested to inhibit FGFR signaling through PLCγ [29].

In summary, FGFR signaling employs a variety of distinct downstream signaling pathways depending on the biological context—most notably, the ERK1/2 MAPK signaling cascade. Of note, it has been suggested that FGFR1 may employ multiple effector pathways (e.g., FRS2, CRK, and PLCγ) converging on the activation of ERK1/2 in multiple cellular contexts (called “additive” signaling), whereas FGFR3 utilizes different pathways (e.g., STAT1 and ERK1/2) to elicit separate cellular responses (called “differential” signaling) [9].

## 3. Oncogenic FGFR Signaling 

In the context of tumorigenesis, in recent years, it has been shown that FGFs play crucial roles in regulating excessive cell growth and angiogenesis. As a matter of fact, they promote hepatocellular carcinoma progression [30], and their serum levels have been suggested to be indicative of recurrent Hepatocellular Carcinoma (HCC) [31] and an independent indicator of lymphatic invasion in patients undergoing the surgical removal of colorectal cancer [32]. Somatic genetic alterations and different patterns of *FGFR* expressions have been observed in many different types of cancers [33,34]. For instance, Helsten et al. demonstrated the presence of *FGFR* alterations in a large cohort of 4853 tumors using next-generation sequencing: *FGFR* mutations were present in 7.1% of malignancies, and these mutations could be categorized as gene amplifications (66%), point mutations (26%), or genomic rearrangements (8%) [35]. In particular, the *FGFR1* gene (located on chromosome 8p11-12) is mutated in many cancers, while the other receptors are not frequently mutated. In breast cancer, for example, *FGFR1* is mutated in ~15% of patients [35,36]. Over the last years, five types of alterations of FGFR signaling have been discovered that promote tumorigenesis: (a) the overexpression of *FGFR* genes (e.g., due to *FGFR* gene amplification) and post-transcriptional alterations that ultimately lead to an increase in FGFR protein levels, (b) the overexpression of FGF ligands in stromal and/or tumor cells, triggering autocrine and/or paracrine pathway activation, (c) *FGFR* point mutations leading to constitutive receptor activation, (d) the expression of FGFR fusion proteins due to genomic translocations causing constitutive activation of the FGFR kinase domain, and (e) the alternative splicing of *FGFR* genes, thereby modifying the *FGFR* isoform and ligand specificity and, hence, the range of FGF ligands capable to bind [6]. As a result of deregulated FGFR signaling, the cells display enhanced proliferation, elevated epithelial-to-mesenchymal transition (EMT), and reduced apoptosis [37].

The amplification of genes encoding for *FGFRs* were first discovered in BC patients in 1991 [38]. In a study involving 1875 breast cancer patients, amplification of the *FGFR1* gene was observed in 10.5% of patients [39]. In addition to FGFR overexpression, elevated levels of FGFs can also trigger angiogenesis. In fact, FGF2 increases the expression of other proangiogenic molecules, such as VEGF and angioprotein-2 [40,41]. Both in vitro and *in vivo* studies indicated a crosstalk between FGF2 and VEGF that affects blood vessel formation and tumor formation [40,42,43]. Furthermore, FGF2 synergizes with platelet-derived growth factor (PDGF)-BB in the promotion of tumor angiogenesis and the formation of scaffolds through the recruitment of pericytes and vascular smooth muscle cells [44]. FGF2 was also shown to upregulate the expression of PDGFR, thereby enhancing the responsiveness to PDGF-BB. On the other hand, PDGF-BB-treated vascular smooth muscle cells upregulate FGFR1, which enhances their responsiveness to FGF2 responsiveness [44]. Of note, the combined inhibition of FGFR and VEGFR reduced angiogenesis and delayed tumor growth in a mouse model of pancreatic cancer, suggesting a potential therapeutic potential of this combination in cancer treatment [45]. Another study showed that blocking PDGFR-β suppressed FGF2 expression, leading to a reduction of cell proliferation and angiogenesis [46]. 

Furthermore, constitutively activating mutations of *FGFR* have been shown to lead to cancer or Crouzon syndrome [47,48,49]. Constitutive activating point mutations trigger conformational changes that cause the receptors to dimerize without the need for ligand binding, followed by trans-phosphorylation of the receptors and activation of the downstream signaling cascade [50]. The activity of the receptors is directed by the transmembrane conformation of the receptor dimers [50]. For instance, it has been reported that FGF1 and FGF2 ligands induce different conformational changes of their receptor, thereby modifying the distance between their intracellular domains and, hence, the number of dimers capable of trans-phosphorylation. This fine-tuning of FGFR signaling can lead to a varying degree of proliferation, apoptosis, EMT, and angiogenesis and may ultimately promote cancer formation [50]. It is important to notice that *FGFR* mutations are not only limited to the kinase domain [35]. Furthermore, in different tissues and different tumor types, distinct *FGFR* mutations have been found. The most well-known activating *FGFR* mutations discovered so far are: (1) N656E and N546K in FGFR1 [51]; (2) N549K, S252W, and P253R in FGFR2 [35,52]; (3) S249C, R248C, G370C, Y373C, R399C, and K650E in FGFR3 [35,53,54]; and (4) N535D, N535K, V550E, and V550L in FGFR4 [35,55].

FGFRs fusion proteins constitute yet another mechanism of oncogenic alterations of FGFR signaling. They account for 8% of FGFR signaling aberrations [35]. Gene fusions leading to FGFR fusion proteins have been discovered most frequently for the family members *FGFR2* and *FGFR3*. For instance, the *FGFR3-TACC3* gene fusion constitutively activates the receptor [56]. Furthermore, *FGFR1* gene fusions have been detected with 11 other genes, such as *BCR*, *FOP*, and *ZNFR3* [57]. Apart from these examples, many other gene fusions have been discovered for *FGFR* family genes [35,58,59].

Interestingly, SNPs in FGFRs can be used to predict the development of premenopausal BC, as evinced by several genome-wide association studies [60,61,62,63]. For instance, Easton et al. showed that *FGFR2* was among the five loci with SNPs significantly associated with breast cancer [60]. Similarly, Stacey et al. described that rs4415084 and rs1094179 were associated with an increased risk of developing BC [61]. In addition, Hunter et al. discovered that four SNPs in *FGFR2* (rs2420946, r1219648, rs2981579, and rs11200014) significantly correlated with malignancy in BC [63]. It has been suggested that such a correlation is due to the presence of a population of tumor-initiating cells, which express high levels of FGFR2 together with GABRA4 and FOXA1. In addition, when *FGFR2* was downregulated using shRNAs, the population of tumor-initiating cells was also significantly downregulated [64].

Several studies have tested FGFR-targeting drugs at various stages of clinical development, and alterations of *FGFRs* can be a predictor for the efficacy of these drugs, as shown for NCP-BGJ398 [65]. For detailed analyses specific on the FGF/FGFR axis in breast cancer, refer to our previous publications [66,67]. Since the focus of this review is on the role of *FGFR* alterations during CDK4/6 inhibitor resistance, the following sections will focus almost exclusively on CDK4/6 inhibition and the biological role of *FGFR* alterations in conferring resistance to CDK4/6-targeting drugs.

## 4. Inhibition of CDK4/6 Proteins for Treatment of Breast Cancer 

The retinoblastoma protein (RB) is an important tumor suppressor that binds to and inhibits the family of E2F transcription factors to prevent unscheduled and excessive cell growth [68]. When the cell is ready for division, RB is phosphorylated and, thereby, inactivated. In several cancers, the growth-inhibitory function of RB is compromised. During transition from the G1 to the S phase of the cell cycle, CDK4 or CDK6 associates with cyclin D family proteins (cyclin D1, D2, or D3). The resulting cyclin D-CDK4/6 complex phosphorylates RB, which, in turn, dissociates from E2F, resulting in cell cycle progression [69]. There are various factors that influence the activity of cyclin D-CDK4/6 to hyperphosphorylate RB, leading to uncontrolled cell proliferation. During the past decade, CDK4/6 inhibitors have gained interest for the treatment of cancer. Currently, there are three CDK4/6 inhibitors in clinical development: palbociclib, abemaciclib, and ribociclib [70]. Their antitumor efficacy, in combination with endocrine therapies, led to their first FDA approval for the treatment of advanced BC in 2015 [71]. In fact, a first-line treatment with palbociclib, abemaciclib, or ribociclib together with aromatase inhibitors significantly prolonged the PFS of patients (from ~15 months to ~25 months) in comparison to aromatase inhibitors alone, as shown in the clinical trials PALOMA-2, MONARCH-3, and MONALEESA-2, respectively [72,73,74]. Moreover, combining palbociclib, abemaciclib, or ribociclib with fulvestrant leads to substantially prolonged PFS compared to fulvestrant alone (as shown in trials PALOMA-3, MONARCH-2, and MONALEESA-3, respectively) when used as a second-line therapy after progression occurs with aromatase inhibitors [75,76,77].

## 5. FGFR Overexpression Correlates with Resistance to CDK4/6 Inhibitors 

For a long time, the loss of RB (encoded by *RB1*) has been considered as the most evident mechanism leading to a resistance to CDK4/6 inhibitor treatment. However, clinical data from the PALOMA-2 trial does not support this hypothesis, because only a small number of patients resistant to CDK4/6 inhibitors were RB-negative (*n* = 51) [72]. FGFRs are involved in many key mechanisms leading to cancer, such as proliferation, differentiation, and cell survival [33]. In addition, as mentioned previously, *FGFRs* are often mutated in BC. Therefore, it is not surprising that the expression or mutation of FGFRs may be associated with disease progression or therapy resistance in BC. For instance, FGFR1 and FGFR2 are not only associated with endocrine resistance but, also, with resistance to CDK4/6 inhibitors [78]. A large genome-scale functional screen regarding the genes involved in endocrine resistance showed that the FGFR, ERBB, insulin receptor, and MAPK pathways are key players in the phenomenon of endocrine resistance. Further comparisons of pretreatment and post-resistance samples revealed that FGFR/FGF axis alterations were acquired under the selective pressure of one ER-directed therapy (tamoxifen, AI, SERDs) [78]. Moreover, there are two activating *FGFR2* mutations (M538I and N550K) that have been identified to be associated with resistance to palbociclib and/or fulvestrant in endocrine-resistant ER^+^ metastatic breast cancer (MBC) patients [78]. In these cases, alternative signaling pathways could possibly drive cancer growth and compensate for the inactivated cyclin D-CDK4/6-RB pathway. For instance, *FGFR1* amplification could lead to the activation of PI3K/AKT and RAS/MEK/ERK signaling pathways in cancer cells that are endocrine therapy-resistant [79]. Clear clinical evidence for the involvement of FGFR signaling in therapy resistance during BC was uncovered from the analysis of data from larger trials. In fact, blood samples of 34 patients from the MONALEESA-2 trial, who progressed after CDK4/6 inhibitor therapy, were further investigated for circulating tumor DNA (ctDNA) using next-generation sequencing [80]. Surprisingly, *FGFR1/2* amplifications or mutations were observed in 41% of the patients (14/34), who progressed after the ribociclib plus fulvestrant treatment. In addition, the occurrence of *FGFR1* amplification in ctDNA samples from patients of the MONALEESA-2 trial correlated with significantly shorter PFS compared to patients with wild-type *FGFR1* [80]. Patients with *FGFR1* amplification treated with ribociclib and letrozole displayed a PFS of 22 months. On the other hand, those patients without *FGFR1* amplification treated with ribociclib and letrozole did not reach the median PFS after 32 months of follow-up, implying a significantly better PFS (HR: 95%; CI: 0.56 (0.36–0.87); *p* = 0.01). To corroborate this, at the ASCO 2019 meeting, O’Leary et al. also showed that *FGFR1* amplification and the *TP53* mutation were significantly correlated with worse survival in a cohort of 521 ER^+^ HER2^−^ MBC patients treated either with palbociclib and fulvestrant or a placebo and fulvestrant [81]. These data further support the notion that *FGFR1* amplification is associated with accelerated disease progression after treatment with CDK4/6 inhibitors and antiestrogens.

Furthermore, involvement of the FGFR signaling pathway during CDK4/6 inhibitor resistance was indicated by a number of very recent reports. For instance, data from 1503 patients in the MONALEESA-2/3/7 trials were presented at the ASCO 2020 meeting. It showed that potential biomarkers for resistance to ribociclib in ctDNA are *FRS2* (altered in >1.87%; *p* = 0.28), *MDM2* (altered in >2%; *p* = 0.06), *PRKCA* (altered; *p* = 0.04), *AKT1* (altered; *p* = 0.03), *BRCA1/2* (altered; *p* = 0.058), and *HER2* (altered in 3.5%; *p* = 0.13) [81]. Therefore, FRS2, HER2, and MDM2 were the most commonly altered biomarkers in relation to ribociclib treatment resistance. With regard to genetic alterations most commonly associated with decreased ribociclib efficacy, the investigators identified alterations in *CHD4* (*p* = 0.0073)*, CDKN2A/B/C* (*p* = 0.046)*,* and *ATM* (*p* = 0.0533) [82]. In addition, Razavi et al. reported an enrichment of alterations in multiple genes among samples from BC patients with CDK4/6 inhibitor resistance, such as *PI3K/AKT*, *RB1*, Hippo signaling genes, *CDKN2A,* and *FGFR1* [83]. The latter gene was amplified in 21% of pretreatment patients and 22% of post-treatment patients. Furthermore, the importance of FGFR/FGF axis deregulation during relapse in the clinical setting of BC was shown by the analysis of patient samples obtained from the MONALEESA-2 trial: Patients with *FGFR1* amplification progressed more rapidly compared to those patients with unaltered *FGFR1* [80].

## 6. Therapeutically Reversing CDK4/6 Inhibitor Resistance by Targeting FGFR

Based on the prevalence of *FGFR* alterations in breast cancer and their relevance during resistance to CDK4/6 inhibition, as discussed above, targeting aberrant FGFR activity may hold clinical value for treating therapy-resistant BC. In a preclinical model of breast cancer using an ER^+^ cell line (MCF-7), the overexpression of *FGFR1* causes a resistance to treatment with fulvestrant and a CDK4/6 inhibitor, either palbociclib or ribociclib [80]. Astonishingly, the acquired resistance was reversed when these cancer cells were subsequently treated with the tyrosine kinase inhibitor (TKI) lucitanib, which targets FGFR, as well as other receptor tyrosine kinases. Most importantly, in a patient-derived xenograft model with *FGFR1*-amplified ER^+^ breast cancer, the addition of the selective FGFR TKI erdafitinib to palbociclib and fulvestrant led to a complete tumor regression in some of the animals. In fact, the triple combination reduced the tumor size by more than 50% after a three-week treatment regimen, while 30% of the xenografts reached a complete response, which was maintained even after treatment was discontinued. Another interesting observation was discovered upon the NanoString expression analysis after treatment with fulvestrant and palbociclib: The expression of most cell cycle genes were reduced, while *FGFR1*, *FGFR2*, *FGFR3,* and the genes involved in MAPK signaling increased [80]. This suggests that the inhibition of CDK4/6, together with endocrine treatment, pushes cells towards an alternative, CDK4/6-independent mechanism to sustain cellular proliferation, such as through the FGFR-MAPK axis. On the contrary, the immunohistochemistry analysis showed that treatment with fulvestrant and the FGFR inhibitor erdafitinib were able to diminish ERα expression and FGFR1 phosphorylation, respectively [80]. Currently, there is an ongoing phase Ib clinical study evaluating a combination treatment with fulvestrant, palbociclib, and the FGFR inhibitor erdafitinib in endocrine-resistant ER^+^ HER2^−^ MBC patients with an amplification of FGFR genes (NCT03238196) (Table 1). Notably, both FGFR signaling and ER transcriptional activity induce the expression of cyclin D1 as a mediator of resistance to endocrine therapies with or without CDK4/6 inhibitors [81,82,83].

Furthermore, Haines et al. showed that, in palbociclib-resistant lung cancer cells (H358 cell line), the expression of G1 cyclins and CDK changed in response to FGFR inhibition [84]. In fact, the treatment with FGFR inhibitor LY2874455 was associated with a decreased expression of CDK6, cyclin D1, and cyclin D3 in these cells. Interestingly, FGFR inhibition exerted the same effect on these cells as MEK and mTOR inhibition. The authors further demonstrated that the secretion of FGF2, also known as basic FGF (bFGF), was capable of promoting resistance to palbociclib and sensitivity to the MEK inhibitor by further enhancing FGFR1 activity. The treatment with FGF2 was shown to be associated with an increase in FGFR1 activation and ERK1/2 phosphorylation, whereas the treatment with an FGF2 neutralizing antibody significantly blocked FGFR1 and ERK1/2 activity. Notably, when these cells were stimulated with FGF2, the cycle profile was not changed by the presence of palbociclib, suggesting that stimulation of the FGF-FGFR axis regulates cell cycle progression irrespective of CDK4/6 activity, and hence, cell cycle is independent from the inhibitory effect of palbociclib [84].

In summary, the data described above strongly suggest that the FGFR pathway is capable of promoting tumor progression even in the presence of CDK4/6 inhibitors and that the addition of FGFR inhibitors is a promising therapeutic avenue to delay or stop tumor progression—in particular, for CDK4/6 inhibitor-resistant BC.

## 7. Other Mechanisms of CDK4/6 Inhibitor Resistance

While the hyperactivation of FGFR represents an important resistance mechanism to the CDK4/6 blockade in vitro and in vivo, there are several other pathways and alterations that confer resistance to CDK4/6 inhibition, either in concert with FGFR or alone. A recent study by Wander et al. reported the whole-exome sequencing of biopsies from ER^+^ BC patients with acquired or intrinsic resistance to CDK4/6 inhibitors [85]. Besides amplification and activating mutations in *FGFR2*, several other genomic alterations likely to mediate resistance were reported. Among them was *RB1* loss and the aberrant activation of HER2 (encoded by *ERBB2*), AKT1, RAS, cyclin E2 (*CCNE2*), and Aurora kinase A (*AURKA*) [85]. Although *RB1* loss is well-documented in mediating resistance to CDK4/6 inhibition [86,87,88], it appears to be a rare event in ER^+^ BC, occurring in only about 5% of patients [89]. In contrast, the hyperactivation of cyclin E-CDK2, another mechanism that has been described early in the search for resistance mechanisms, seems to be a frequent event in patient tumors. Cyclin E-CDK2 mediates G1-S progression of the cell cycle in concert with cyclin D-CDK4/6 [90]. Similar to cyclin D-CDK4/6, cyclin E-CDK2 complexes phosphorylate RB, thereby allowing E2F transcriptional activity and cell cycle progression. As cyclin E-CDK2 acts downstream of cyclin D-CDK4/6, the hyperactivation of cyclin E-CDK2 renders CDK4/6 inhibition ineffective and allows cell cycle progression despite the CDK4/6 blockade. In line with this, an analysis of the data from the PALOMA-3 clinical trial (palbociclib in combination with fulvestrant in patients with advanced ER^+^ BC) confirmed the important role of cyclin E during the response to CDK4/6 inhibitors. Patients with a high expression of cyclin E1 mRNA displayed a significantly shorter PFS compared to patients with low levels of cyclin E1 mRNA (7.6 months vs. 14.1 months) [91]. The hyperactivation of cyclin E-CDK2 can be mediated by several mechanisms. The amplification of *CCNE1* and *CCNE2*, the genes encoding cyclin E1 and cyclin E2, leads to significantly increased cyclin E protein levels [85,92]. Furthermore, the downregulation of the endogenous CDK2 inhibitor proteins p21CIP1 and p27KIP1 leads to the increased activity of cyclin E-CDK2 complexes and has been shown to occur in CDK4/6 inhibitor-resistant BC cells [93]. A recent study by Costa et al. showed that *PTEN* loss and the subsequent activation of AKT signaling causes the exclusion of *p27KIP1* from the nucleus, resulting in the reduced p27KIP1-mediated inhibition of CDK2 and, hence, enhanced cyclin E-CDK2 activity. Importantly, the cotreatment with a CDK4/6 inhibitor and an inhibitor of AKT signaling restored the sensitivity to CDK4/6 inhibition in vitro and in vivo [94]. 

Moreover, Jansen et al. demonstrated that increased levels of PDK1 can drive a resistance to CDK4/6 inhibition [95]. In this case, the aberrant activation of downstream AKT signaling results in increased expression of cyclins E, A, and D1, as well as increased activation of CDK2. Importantly, the addition of a PDK1 inhibitor to CDK4/6 inhibitor treatment reversed the resistance in ER^+^ breast cancer cell lines, and both PDK inhibition, as well as PI3K inhibition, potentiated the effect of a CDK4/6 blockade in xenograft models. The inhibition of AKT signaling therefore represents a promising addition to CDK4/6 inhibition in the clinic to prevent or delay resistance [95].

Another mechanism of CDK4/6 inhibitor resistance is the overexpression of CDK6 [87,96,97]. While, in some cases, this is triggered by *CDK6* gene amplification [96], the upregulation of CDK6 was recently shown to be linked to *FAT1* loss, causing activation of the HIPPO pathway [87]. In line with this, patients with FAT1-deficient tumors displayed a significantly shortened PFS compared to patients with FAT1-proficient tumors. Furthermore, a study by Cornell et al. reported an increased expression of CDK6 being elicited by suppression of the TGF-β pathway via exosomal miRNA 432-5p [98]. 

Finally, in triple-negative breast cancer (TNBC), an increased number of tumor cell lysosomes has been shown to mediate the intrinsic resistance to CDK4/6 inhibition [98]. Driven by their weak base properties, CDK4/6 inhibitors are sequestered into the enlarged lysosomal compartment and, therefore, are hindered from reaching their target, i.e., cyclin D-CDK4/6 complexes in the nucleus. Importantly, a cotreatment with lysosomotropic agents such as chloroquine and azithromycin rendered TNBC cells fully sensitive to CDK4/6 inhibition [98].

## 8. Conclusions

*FGFR* alterations occur frequently in breast cancer patients. Importantly, aberrant activation of the FGFR/FGF axis is commonly involved in endocrine resistance or resistance to CDK4/6 inhibitors during the treatment of BC. Mechanistically, this may occur through cellular rewiring of ER signaling and the activation of several pathways, such as the ERK1/2 MAPK signaling pathway. Although FGFR signaling induces cyclin D1, it is unlikely that cyclin D1 would be involved in the development of CDK4/6 inhibitor resistance. As a matter of fact, earlier data shed light on this matter, showing that there is a lack of association between cyclin D1 amplification with the survival of patients treated with the CDK4/6 inhibitor palbociclib (in the PALOMA-1/2 trials). Taken together, these data encourage the support for clinical trials using combinations of ER, CDK4/6, and FGFR or ERK antagonists in therapy-resistant BC patients. Since FGFR1 is among the most common mutations in BC patients, a concomitant treatment with the inhibitors of ER, CDK4/6, and FGFR1 could also be a valid first-line treatment alternative to avoiding the emergence of therapy resistance. Additionally, we herein discussed that various other mechanisms were identified that confer CDK4/6 inhibitor resistance. With the advent of molecular precision medicine, it will be both important and feasible to identify the specific resistance mechanism for each patient to select patients that will most likely benefit from an additional line of targeted therapy, such as FGFR inhibition.

## Figures and Tables

**Figure 1 cells-10-00293-f001:**
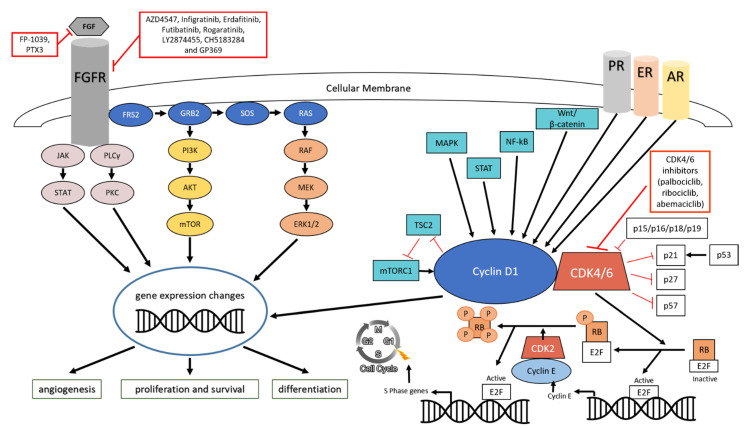
Fibroblast growth factor-FGF receptor (FGF-FGFR) and cyclin D-cyclin-dependent kinase (CDK)4/6 pathways in breast cancer. The molecular components of these signaling pathway and their connections are indicated, together with the available inhibitors that target FGF, FGFR, or CDK4/6.

**Table 1 cells-10-00293-t001:** Clinical trials using selective fibroblast growth factor receptor (FGFR) inhibitors in breast cancer.

Identifier	Phase	Administrated Drug and Setting	Primary Endpoint	Status
NCT03238196	I	Erdafitinib; Palbociclib; Fulvestrant	AE	Recruiting
NCT04483505	I	Rogaratinib + palbociclib + fulvestrant	Recommended Phase 2 Dose; AE	Recruiting
NCT03344536	I/II	Fulvestrant; Debio 1347	DLT (phase I); Best ORR (phase II)	Active, not recruiting
NCT04504331	I	Infigratinib; Tamoxifen; Omnipaque 350; Iopamidol	DLT	Recruiting
NCT04024436	II	TAS-120; Fulvestrant	ORR, CBR, PFS	Recruiting
NCT02299999	II	AZD2014; AZD4547; AZD5363; AZD8931; Selumetinib; Vandetanib; Bicalutamide; Olaparib; Anthracyclines; Taxanes; Cyclophosphamide; DNA intercalators; Methotrexate; vinca alkaloids; Platinum based chemotherapies; Bevacizumab; Mitomycin C; Eribulin; MEDI4736	PFS	Active, not recruiting
NCT02465060	II	Adavosertib; Afatinib; Afatinib Dimaleate; Binimetinib; Capivasertib; Copanlisib; Copanlisib Hydrochloride; Crizotinib; Dabrafenib; Dabrafenib Mesylate; Dasatinib; Defactinib; Defactinib Hydrochloride; Erdafitinib; FGFR Inhibitor AZD4547; Ipatasertib; Larotrectinib; Larotrectinib Sulfate; Nivolumab; Osimertinib; Palbociclib; Pertuzumab; GSK2636771; Sapanisertib; Sunitinib Malate; Taselisib; Trametinib; Trastuzumab; Trastuzumab Emtansine; Ulixertinib; Vismodegib	ORR	Recruiting
NCT04125693	II	Rogaratinib (BAY1163877); Combination drug	TEAE	Enrolling by invitation
NCT01791985	IIa	AZD4547 + anastrozole or letrozole	Safety and tolerability; Change in tumor size at 12 weeks	Completed
NCT01202591	I/II	AZD4547 + exemestane; AZD4547 + fulvestrant; Placebo + fulvestrant	Safety and Tolerability	Completed

AE, Adverse Events; DLT, Dose-Limiting Toxicity; ORR, Overall Response Rate; CBR, Clinical Benefit Rate; PFS, Progression-free survival; and TEAE, Treatment-emergent adverse events.

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
