# Peer review of "Targeting Aberrant FGFR Signaling to Overcome CDK4/6 Inhibitor Resistance in Breast Cancer"

_cells, 2021, doi:10.3390/cells10020293_

Round 1

Reviewer 1 Report

Authors Sobhani et al present a review on the relevance of FGFR and breast cancer drug resistance. the focus appears to be on endocrine resistant in ER+ breast cancer. This review provides new information and offers a potential use of adjuvant therapy. It is well written and clear. If would benefit from minor alterations that would provide focus and clarify. Specifically to effects of FGFR expression, drug resistance and the type of endocrine therapy used.

  1. Page 3 they outline the type of ligands involved in FGF signaling and highlight which ligands are involved in endocrine signaling or table or figure outlining the different ligands and their function would be beneficial for readers. They refer to ligands that can act in an endocrine function, authors should highlight if particular ligands have different impacts on breast cancers. Do all ligands induce/regulate the same pathways? Differences and relevance to breast cancer should be highlighted.
  2. Furthermore, authors highlight where mutations, SNPs and amplifications are observed in BC trials where endocrine resistance and drug resistance occur. As endocrine therapies have different modes of action, authors should highlight were each endocrine drug and subsequent resistance occurs with alterations to FGFR (mutation, SNP, amplifications). This would go well prior to information in table 1.
  3. Figure 1 seems like an oversimplification of ER singling, ER also cross talks with ERK and AKT-which is activated by FGFR, how does this play in with drug action?
  4. Authors state that “Further comparison of pre-treatment and post-resistance samples revealed that FGFR/FGF axis alterations were acquired under the 268 selective pressure of ER-directed therapy [77].” They should highlight what the therapy is as endocrine therapies act through different mechanisms.
  5. Throughout authors should define what type of therapy resistance they are referring to (chemotherapy, endocrine therapy-tamoxifen, ICI, aromatase inhibitor). For instance, page 7 line 276 what type of drug resistance is being referred to? There is mention of aromatase inhibitors on line 285 but these should be highlighted prior to this.

Author Response

Reviewer 1

Date of this review: 19 Jan 2021 02:15:50

Comments and Suggestions for Authors:

Authors Sobhani et al present a review on the relevance of FGFR and breast cancer drug resistance. the focus appears to be on endocrine resistant in ER+ breast cancer. This review provides new information and offers a potential use of adjuvant therapy. It is well written and clear. If would benefit from minor alterations that would provide focus and clarify. Specifically to effects of FGFR expression, drug resistance and the type of endocrine therapy used.

  1. Page 3 they outline the type of ligands involved in FGF signaling and highlight which ligands are involved in endocrine signaling or table or figure outlining the different ligands and their function would be beneficial for readers.

We added the right FGFs binding to FGFRs into the figure.

They refer to ligands that can act in an endocrine function, authors should highlight if particular ligands have different impacts on breast cancers. Do all ligands induce/regulate the same pathways? Differences and relevance to breast cancer should be highlighted.

In this paper we have explained everything known in literature about FGFR and CDK4/6 resistance. For more comprehensive analyses on FGF/FGFRs axis in breast cancer please refer to our highly cited publications:

  1. Sobhani, N.; Ianza, A.; D’Angelo, A.; Roviello, G.; Giudici, F.; Bortul, M.; Zanconati, F.; Bottin, C.; Generali, D.; D’Angelo, A.; et al. Current Status of Fibroblast Growth Factor Receptor-Targeted Therapies in Breast Cancer. Cells 2018, 7, 76, doi:10.3390/cells7070076.
  2. Sobhani, N.; Fan, C.; O. Flores-Villanueva, P.; Generali, D.; Li, Y. The Fibroblast Growth Factor Receptors in Breast Cancer: from Oncogenesis to Better Treatments. Int. J. Mol. Sci. 2020, 21, 2011, doi:10.3390/ijms21062011.

  1. Furthermore, authors highlight where mutations, SNPs and amplifications are observed in BC trials where endocrine resistance and drug resistance occur. As endocrine therapies have different modes of action, authors should highlight were each endocrine drug and subsequent resistance occurs with alterations to FGFR (mutation, SNP, amplifications). This would go well prior to information in table 1.

The focus of our review is CDK4/6inhibitors resistance and FGFR mutations. To help readers we summarized endocrine treatments with a more general classification.

  1. Figure 1 seems like an oversimplification of ER singling, ER also cross talks with ERK and AKT-which is activated by FGFR, how does this play in with drug action?

We added new lines of interactions showing ER cross talks with ERK and AKT, which are activated through the FGFRs. It shows the interconnections. The drugs inhibiting CDK4/6 block CDK4/6 that usually makes a complex with cyclin D1 to exert its function. ER, PR, AR all interact with cyclin D1. However, as we stated in the conclusions cyclin D1 does not have a role in conferring resistance to CDK4/6 inhibitors. In fact, we rephrased one of the conclusive remarks saying “Although FGFR signaling induces cyclin D1, it is unlikely that cyclin D1 would be involved in the development of CDK4/6 inhibitor resistance. As a matter of fact, earlier data shed light on this matter showing that there is a lack of association between cyclin D1 amplification with survival of patients treated with CDK4/6 inhibitor palbociclib (is PALOMA-1/2 trials).”  

  1. Authors state that “Further comparison of pre-treatment and post-resistance samples revealed that FGFR/FGF axis alterations were acquired under the 268 selective pressure of ER-directed therapy [77].” They should highlight what the therapy is as endocrine therapies act through different mechanisms.

We added the information: “one ER-directed therapy (tamoxifen, AI, SERDs)”

  1. Throughout authors should define what type of therapy resistance they are referring to (chemotherapy, endocrine therapy-tamoxifen, ICI, aromatase inhibitor). For instance, page 7 line 276 what type of drug resistance is being referred to? There is mention of aromatase inhibitors on line 285 but these should be highlighted prior to this.

We fixed the two endocrine treatments on page 7. The focus of our review is CDK4/6inhibitors resistance and FGFR mutations. To help readers we summarized endocrine treatments with a more general classification. Additionally, CDK4/6 inhibitors block CDK4/6 that usually makes a complex with cyclin D1 to exert its function. ER interacts with cyclin D1. However, as we stated in the conclusions cyclin D1 does not play a significant role in conferring resistance to CDK4/6 inhibitors. In fact, we rephrased one of the conclusive remarks saying “Although FGFR signaling induces cyclin D1, it is unlikely that cyclin D1 would be involved in the development of CDK4/6 inhibitor resistance. As a matter of fact, earlier data shed light on this matter showing that there is a lack of association between cyclin D1 amplification with survival of patients treated with CDK4/6 inhibitor palbociclib (is PALOMA-1/2 trials).” 

Reviewer 2 Report

Sobhani et al. present a review summarizing the potential of targeting FGFR signaling to overcome CDK4/6 inhibitor resistance in breast cancer. This is a timely review and includes a comprehensive body of work. It is also well structured and has a logical flow. There are some suggestions and minor corrections:

  1. In the abstract (Line 20), the authors stated that “CDK4/6 inhibitors trigger dephosphorylation of Rb…”. Is this statement accurate, because it potentially implies a phosphatase acting on Rb upon inhibitor treatment?
  2. For the introduction (~Lines 60-62), there seems to be a gap in justifying the rationale for focusing on FGFR signaling. This foundation should be laid out in the introduction or at least the authors’ opinion for focusing on this pathway should be included.
  3. Line 120, spelling error for GRB2
  4. Line 228, spelling error for tested
  5. Line 270, structure of sentence needs to be revised.
  6. Line 422, structure of sentence needs to be revised.

Author Response

Reviewer 2

Date of this review: 19 Jan 2021 15:27:58

Comments and Suggestions for Authors

Sobhani et al. present a review summarizing the potential of targeting FGFR signaling to overcome CDK4/6 inhibitor resistance in breast cancer. This is a timely review and includes a comprehensive body of work. It is also well structured and has a logical flow. There are some suggestions and minor corrections:

  1. In the abstract (Line 20), the authors stated that “CDK4/6 inhibitors trigger dephosphorylation of Rb…”. Is this statement accurate, because it potentially implies a phosphatase acting on Rb upon inhibitor treatment?

That is correct. Ultimately the CDK4/6 inhibitors do that, but indirectly. We adjusted the abstract” CDK4/6 inhibitors indirectly trigger dephosphorylation of retinoblastoma tumor suppressor protein by blocking CDK4/6, thereby blocking cell cycle transition from G1 to S phase”

  1. For the introduction (~Lines 60-62), there seems to be a gap in justifying the rationale for focusing on FGFR signaling. This foundation should be laid out in the introduction or at least the authors’ opinion for focusing on this pathway should be included.

Thank you. We corrected it.

  1. Line 120, spelling error for GRB2

Thank you. We corrected it.

  1. Line 228, spelling error for tested

Thank you. We corrected it.

  1. Line 270, structure of sentence needs to be revised.

Thank you. We revised it.

  1. Line 422, structure of sentence needs to be revised.

Thank you. We revised it.